# Lessons in a Green School Environment and in the Classroom: Effects on Students’ Cognitive Functioning and Affect

**DOI:** 10.3390/ijerph192416823

**Published:** 2022-12-15

**Authors:** Lucia Mason, Lucia Manzione, Angelica Ronconi, Francesca Pazzaglia

**Affiliations:** 1Department of Developmental Psychology and Socialization, University of Padova, 35122 Padova, Italy; 2Department of General Psychology, University of Padova, 35131 Padova, Italy

**Keywords:** green environment, classroom environment, school lessons, attention, math performance, positive affect, negative affect, emotional difficulties

## Abstract

The positive impact of short-term exposure to nature during a green recess in a school day is documented in the literature. In this study we investigated cognitive, academic, and affective effects of a single contact with nature during a regular school lesson in the greenness, compared to an usual classroom lesson, on young students in second and third grades (N = 65). In a within-subjects design, for the cognitive effects we examined children’s (a) selective and sustained attention and (b) math calculation performance in common school tasks. For affective effects we considered (c) their positive and negative mood and (d) the perception of environmental restorativeness. Findings revealed that after a single lesson taught in the green school garden, children had greater selective attention and math calculation performance in two tasks than after a similar lesson in the classroom environment. Moreover, children with higher self-reported emotional difficulties showed greater selective attention and reported a statistically significant increase in positive affect and a tendency to a significant decrease in negative affect after the lesson in the greenness than in the classroom. Students also perceived the green space as more restorative than the classroom environment. Results are discussed against theories on the benefits of exposure to natural environments, highlighting the theoretical and practical significance of the study.

## 1. Introduction

The benefits of long-term exposure to nature have been widely documented in the literature, in particular regarding environmental and architectural research. Systematic reviews have reported the positive effects of green environments on adults’ physical and mental health [1], cognition of people at various ages including childhood and adolescence [2,3], behavior, cognition, and emotion of young people up to 18 years [4], health and wellbeing in children [5], and school achievement [6].

Such benefits are associated with two important characteristics of the contact with greenness. First, it is a passive contact, that is, the natural environment is not used or incorporated in people’s activities as it may happen, for instance, when they are involved in gardening and horticulture, or when for biological and environmental education students are involved in observing small animals of the soil. Second, the literature refers to a long-term contact with green areas. For example, a systematic review on outdoor education programs has documented their effectiveness for students’ academic achievement and social and health dimensions [7]. Similarly, another systematic review has reported various dimensions of psychological benefits of attending forest schools for preschoolers [8]. Recently, a study has also indicated that teaching regular lessons not in the classroom but in nature over a school term led eighth graders to spending more time on-task, an effect that lasted for weeks, although higher engagement did not result in better grades for several weeks [9].

However, a systematic review also reported positive effects for cognitive functioning that are related to short breaks (e.g., 15–20 min) as green recesses during a school day in students of primary and secondary school, and college [10]. Based on the current literature, therefore, we know that both long-term passive contact with nature surrounding schools [6] and houses [11], or traditional school lessons in green areas are beneficial for cognitive and academic functioning [7]. We also know that short-term passive contacts with natural environments during breaks positively impact cognitive functioning [10]. In light of previous research, we wondered if even the effects of a single school lesson in a green space would be superior than those of a lesson in the usual classroom not only for children’s cognitive and academic functioning, but also for their emotional wellbeing and perception of the restorative quality of the environment where the lesson took place. It is certainly practically easier and feasible to teach outdoor once in a while than over a long period of time in ordinary schools.

### 1.1. Why Is Exposure to Nature Beneficial?

Two theories have been developed to explain the mechanisms underlying the positive effects of exposure to green environments: one is the attention restoration theory (ART) [12] and the other is the stress reduction theory (SRT) [13]. According to ART, when we are visually exposed to nature, we can restore or regenerate our directed attention that is top-down, voluntary, and requires mental effort. Thus, directed attention is subject to depletion after an intense cognitive engagement as to stay focused on what may not be interesting, we need to use inhibitory control of distractions [12,14]. In contrast, involuntary attention that activates, for instance, when something interesting or strange happens, is bottom-up and automatically attracts us at no expenses of our mental resources. Therefore, only voluntary attention needs to be restored and a way to allow it to rest is to rely on involuntary attention.

Natural environments (e.g., forests, parks, gardens) are rich of stimuli that softly capture involuntary attention playing down voluntary attention, that is, greenness does not overwhelm our attentional system [12,14]. That is, natural environments minimize voluntary attention with irrelevant stimuli being ignored more readily through the mechanism of inhibitory control of distractions. As a consequence, after exposure to nature our capacity to deliberately focus attention on a task is restored. In contrast, urban environments have very low, or not at all, restorative function as their stimuli require directed attention as in case of a red traffic light. To sum up, the concurrent combination of attracting involuntary attention and reducing the demand of direct attention is what characterizes restorative environments such as green spaces [14].

According to the stress reduction theory (SRT), the positive impact of nature on affect and wellbeing is related to the decrease of stress which is a psychophysiological condition of response to environmental stimuli that represent a threat or challenge [13,15]. Psychologically, stress is accompanied by negative emotions such as anger, anxiety, or sadness as a consequence of cognitive appraisal of a situation. Physiologically, stress reflects in various systems of our body, like the cardiovascular (e.g., increased heart rate), which activates to respond to environmental demands, consuming resources especially if stress is protracted. Behaviorally, stress interferes with cognitive functioning leading to a decrease of performance [13]. SRT states that the restorative effect of green environments, that is a reduction in physiological arousal and activation of the parasympathetic nervous system, is related to the role that nature played during human evolution by offering protection and safety [15]. Of note is that in SRT, recovery from psychophysiological stress leads to enhancing well-being [13,16].

To synthesize, both ART and SRT posit that green areas are restorative, although the two theories differ for the mechanisms through which restoration occurs. According to ART, the mechanism is restoration of mental fatigue, whereas according to SRT, it is reduction of stress when individuals are exposed to nature [17]. ART and SRT are not conflicting as, on the one hand, low level of directed voluntary attention can be associated with stress and negative affective states; on the other hand, reduction of stress can be associated with an increase in directed attention and positive affect [18,19].

### 1.2. Cognitive Benefits of Exposure to Nature in the School Context

ART is the basis of most investigations that document the positive effects of experiences with nature on cognitive functioning e.g., [20]. When considering the school context, there are studies focused on long-term exposure to greenness, that is an accumulated effect on students’ achievement [21,22,23,24]. For instance, Li et al. [24] indicated that within a radius of a mile from high schools in Illinois (US), the density of tree cover was positively related to ACT scores–reflecting the school average of the composite score of four subjects, English, mathematics, reading, and science–and college readiness after controlling for student, school, teacher, and location variables contributing to academic achievement. Flouri et al. [11] indicated that children living in greener urban areas have better spatial working memory, a strong correlate of academic achievement e.g., [25]. However, it should be noted that there is also scientific research that reports no relationship between green spaces and students’ achievement, or even a negative correlation e.g., [26,27,28]. Explanations for the mixed outcomes refer to the lack of control for the quality of greenness or the actual exposure to nature for each student [26], the use of multiple ways to measure natural environments by means of remote sensors at different resolutions, and the disregard of population density as a measure of urbanicity [27]. To distinguish between tree and other types of green cover as well as taking into account socio-economic disadvantage not only as a confound but as a mediator, future studies have been proposed on the topic [28].

There are also studies that grounded on ART to investigate the effects of nature on short breaks during a school or study day [10]. For instance, Amicone et al. [29] documented that fourth and fifth graders had higher scores for selective and sustain attention, and working memory after a green break of 30′ in the morning in the school garden (Study 1) or a green break in the afternoon after lunch (Study 2) compared to outdoor breaks in the school built courtyard. Indeed, students can “reload batteries” [30] by a short recess in a green area. Attentional benefits did not emerge in studies in which children were not induced intense mental fatigue before a break in nature or when the used test was suboptimal for detecting the aspects of attention that can be restored after rest in a green environment [31].

In the school context, attention is one of the essential functions to be activated in performing learning activities [32]. Focused and sustained attention is needed to listen to a teacher’s explanation, to read and write a text, or to compute arithmetical operations and solve problems, just to mention few important school tasks e.g., [33]. Attention is at the basis of executive functioning and self-regulation tasks [12], which contribute to academic achievement [34]. With respect to this, it has been stated that few constructs may have a more direct impact on children’s academic achievement than their ability to pay attention in the classroom [35].

### 1.3. Emotional Benefits of Exposure to Nature in the School Context

SRT is also supported by several investigations that document decrease of negative affect and physiological stress responses in people of different ages when they have contact with nature [5,36,37]. For example, Flouri et al. [38] revealed that access to garden and use of parks and playgrounds were associated with children’s (aged 3–5) fewer conduct, peer, and hyperactivity problems as reported by their parents. Moreover, poor children who lived in greener urban neighborhoods had fewer emotional problems than those who lived in less green neighborhoods. A systematic review regarding the relationships between brain activity, mood, and environment by Norwood et al. [39] evidenced that green environments were associated with more comfortable and restorative feeling than the urban ones which were more related to effortful attention. Regarding the school context, it has also been documented that making the schoolyard greener led students’ to perceiving it as more restorative, reduced their physiological stress, and increased their psychological wellbeing [40]. Roe and Aspinall [41] found greater positive change in affect regarding energy, stress, anger, and hedonic tone of young adolescents with “poor” and “good” behavior after a day in an outdoor educational setting (forest school) compared to an indoor setting. Li and Sullivan [19] also reported, among other benefits, reduction in physiological stress in high school students who were exposed to nature for 45 min only through a window green view. Moreover, Han [18] indicated that college students who walked and performed low or moderate physical activity for 15 min in a natural environment reported lower levels of fatigue and nervousness than those who walked and exercised in a built environment.

Again, Mygind et al. [31] documented stress reduction in primary school students (aged 10–12) after spending one hour in a natural environment, as revealed by physiological parameters such as tonic and phasic vagal tone indicating increased activity of the parasympathetic system and decreased activity of the sympathetic system. Wallner et al. [30] also found that high-school students’ immediate and sustained wellbeing was higher after an one-hour stay in a forest space compared to other different urban green spaces. In a prospective longitudinal study, Dettweiler et al. [42] also reported that children (aged 11 years) who were taught in a natural environment had a larger decrease of salivary cortisol levels, that is the stress hormone, over a school day than children who were regularly taught in school, regardless the time of the year and levels of physical activity. Furthermore, there is evidence of fourth and sixth graders’ more efficient regulation of biological-stress reactivity and decrease of cortisol during the day associated with light physical activity in a forest environment [43]. Importantly, stress [44], negative mood [45], and negative emotions [46,47] can have a strong adverse effect on academic performance.

### 1.4. The Current Study

From the aforementioned studies, we know that even short-term passive contacts with nature during a break have positive effects on cognitive functioning and emotional wellbeing [10]. We also know that lessons over a school term produced more engagement in students but not better grades [9]. However, there are no studies that investigated whether a single passive and relative short exposure to nature during a school lesson, not a break, is less mentally fatiguing, that is, depletes the capacity-limited attentional system less [14], and induces more wellbeing than a lesson in the indoor classroom environment [31]. To fill in this gap and extend current knowledge, starting from a simple but rigorous and practically relevant research design, we considered it worth investigating the impact of a single school lesson taught in a green space compared to a lesson in the usual classroom environment on both cognition and affect. More specifically, for cognitive functioning we examined not only children’s attention performance in a typical lab task, as in most previous studies, but also calculation performance in typical school tasks that require selective and sustained attention. For affect we examined children’s positive and negative affective states, and their perception of the restorative quality of outdoor and indoor environments.

Furthermore, we considered the role of students’ self-reported emotional difficulties as this individual difference could moderate the effects of greenness. Emotional difficulties refer to negative feelings and emotions that are directed inward, such as sadness, anxiety, and depression e.g., [48]. With respect to this, studies indicated the positive role of urban neighborhood green space in children’s emotional resilience [11]. Research has also shown that children were less distracted from negative emotional materials when presented with outdoor than indoor background stimuli [49]. Students with higher or lower emotional difficulties may therefore benefit from exposure to nature differently. Four research questions guided the study:

RQ1: Do students’ selective and sustained attention, as measured in a typical lab task, differ in relation to the physical environment where a lesson was taught?

RQ2: Do students’ academic performance, as measured in typical math calculation tasks, also differ in relation to the physical environment where the lesson was taught?

RQ3: Do students’ affective state after the lesson differ in relation to the physical environment where the lesson was taught?

RQ4: Do students perceive differently the restorativeness of the physical environment where the lesson was taught?

Importantly, for each research question we also asked whether students’ self-reported emotional difficulties moderated the relationship between the environment and the outcome variable. Based on research grounded on both ART and SRT regarding the benefits of exposure to nature, we hypothesized the positive effect of the green environment for attention (Hypothesis 1), math calculation (Hypothesis 2), affective state (Hypothesis 3), and perception of environmental restorativeness (Hypothesis 4). That is, greater performances after the lesson in the greenness than in the classroom, as well as greater affective state and perception of the green environment as nature distracts less by irrelevant stimuli, thus depletes the limited attentional resources less [23,50] and induces more emotional wellbeing [4,51] compared to the classroom environment. Moreover, we explored the moderating role of emotional difficulties in the relationship between the environment and the dependent variables. As green spaces are beneficial from multiple perspectives [11,13,23,31], students with higher emotional difficulties might be more “sensitive” to the positive effects, both cognitive and emotional. As a consequence, their cognitive and academic functioning, as well as their affective states, are greater after a lesson in the greenness than indoor classroom.

## 2. Methods

### 2.1. Research Design

We used a within-person design to investigate the cognitive and affective effects of two regular school lessons, one taught outdoor in the green school garden and the other indoor in the classroom.

### 2.2. Participants

We involved 65 children in the second and third grades from four classes of two primary schools located in northern Italy (F = 30; *M*_age_ = 8.25, *SD* = 0.43) during the month of May 2022. Of these children, 63 had Italian as their native language. The two non-Italian children were from the Philippines and had sufficient language knowledge to take part in the study. Students participated voluntarily and written informed consent was obtained from their parents. Permissions were first given by school principals and teachers. The study was approved by the university ethics committee.

### 2.3. Measures

#### 2.3.1. Attention Task

As a measure of selective and sustained attention, we used the Bells test [52], a paper-and-pencil standardized test consisting of four sheets, each containing figures of 35 black bells among 280 distracting stimuli (e.g., trees, horses, houses). Children are asked to find as many bells as possible in two minutes for each sheet (maximum score for each sheet = 35). Score of selective attention is the number of bells marked on the first sheet. Score of sustained attention is the number of bells marked on the fourth sheet. The same test was used after the indoor and outdoor lesson to compare children’s performances in the two environmental conditions. However, it should be noted that it is highly unlikely that children can remember the exact position of the bells among the distracting stimuli in each sheet.

#### 2.3.2. Math Calculation Tasks

Two calculation tasks for the grade levels were used. One task involved performing as many of 64 row additions and subtractions as possible in a shorter time frame (three minutes); half were additions and half subtractions (maximum score = 64). The other task involved performing eight column operations (additions, subtractions, multiplications, and divisions) in a longer time frame (five minutes; maximum score = 8). Of note is that we were not interested in new learning related to the lessons, so we used two processing speed math tasks that all students were supposed to be able to perform. The operations included in the two math tasks where not the same in the indoor and outdoor sessions to avoid any possible repetition effect in this case, but they were similar for the level of difficulty.

#### 2.3.3. Emotional Difficulties

Self-reported emotional difficulties were assessed using the scale included in the Strengths and Difficulties Questionnaire (SDQ) [53]. It comprises five items (e.g., “I am often unhappy, depressed or tearful”) to be rated on a three-point scale (0 = not true; 2 = certainly true; McDonald’s ω = 0.65). Although the behavioral screening questionnaire was originally validated on a sample of parents and teachers of children drawn from dental and psychiatric clinics, the child version of the instrument has also been used with samples of typically developing fifth graders [54]. Based on our participants’ literacy and comprehension ability, and on a research, not clinic purpose, we used with younger children the child version of the scale included in the SDQ. Importantly, each question was read aloud to the class, and any difficult words or phrases were explained and answers were provided when children asked clarifications.

#### 2.3.4. Affective State

We used a short version of the Positive and Negative Affect Schedule (PANAS) [55] that assesses the current mood. Children rated a list of six adjectives describing how they feel at the time on a five-point scale (0 = not at all; 4 = extremely). The list of emotional terms can be grouped into positive (e.g., calm; 3 items) and negative (e.g., nervous; 3 items) affect.

#### 2.3.5. Perceived Restorativeness

We used a short version ([56] of the original self-report scale (PRS) [57], like in Amicone et al. [29]. The 11-point scale (0 = not at all; 10 = completely) comprises four items (e.g., “This place is fascinating”). As measured by McDonald’s ω, reliability was 0.69 for the classroom and 0.79 for the green school garden.

### 2.4. Procedure

For each class, the data collection took place over three different sessions. In the first session students generated their own code for anonymity of the tasks and completed the emotional difficulties scale. The session lasted about 30 min. The second session took place a week later in the green garden or in the classroom. Before a 45-min lesson, children reported their affective state. At the end of the lesson, they reported again their affective state and executed the shorter-term math task, followed by the attention and longer-term math tasks. Finally, children also completed the scale for perceived restorativeness of the environment in which they were. The third session, a week later, was similar to the previous and children completed the same tasks, but it took place in the other environment. Of note, about half children had the second session in the greenness and the third in the classroom and vice versa.

Importantly, the two lessons were taught by the same class teachers and were similar for content difficulty about grammar or science. Teachers were instructed not to interact with the natural environment to ensure that potential differences would be ascribed to passive exposure to nature and not to that interaction. During the green lesson, in one school garden students were seated in circle on wooden logs which served as seats, both for listening to the lesson and to execute the tasks. In the other school garden, they were seated on the grass, directly in contact with nature. In both gardens there were trees close to the children (see Figure 1). The indoor environment was the usual school classroom. The approximate length of the second and third sessions was between 1 h and 1 h and 10 min.

### 2.5. Analytical Plan

We performed linear mixed models (LMM) that allow to simultaneously consider all variables that potentially contribute to the nested structure of the data. These variables included not only the fixed-effect factors (i.e., environment, emotional problems, and their interaction), but also the random effects of participant and class. For each of the outcomes, class and student variables were therefore used as random effects. Of note is that the class was not included in the random effects for negative affect as the variance explained was really close to zero. Moreover, as classes were nested in only two schools, school was not considered as a random effect. To ensure unbiased variance estimations, we computed parameters using the restricted maximum likelihood procedure [58]. In addition, both 95% confidence interval and statistical significance are reported for each parameter. The goodness of fit of the mixed-effects model is indicated by marginal and conditional *R*^2^, which represent the variance explained by the fixed and both the fixed and random factors, respectively [59], and the intra-class correlation coefficient (ICC). In addition, we report the variance explained by the students (τ_00-ID_) and the class (τ_00-Class_), and the variance of the residuals (σ^2^). All statistical analyses were performed with the R software [60], version 4.1.1, using the “lmer” function from the package “lme4” (version 1.1-29) for LMMs fitting, the “sim_slopes” function from the package “interaction” (version 1.1.5) for the slope analyses, and the packages “sjPlot” and “ggplot2” for the figures.

## 3. Results

Data were first tested for normal distribution. The measured variables were normally distributed, except the negative affect before and after the lesson in each of the two environments. The measures of negative affect were therefore log-transformed to improve the validity of the statistical analyses. We present the results organized by the four research questions. Table 1 reports the descriptive statistics for all variables.

### 3.1. RQ1: Effect of the Environment on Attention Performance

For selective attention, LMM revealed the main effect of the environment and the interactive effect of environment * emotional difficulties (Table 2, panel A). The model explained 43% of the variance (marginal *R*^2^ = 20% explained by the fixed effects). Specifically, children had higher scores after the green lesson.

The slope analysis showed that both children with lower (*B* = 3.20, *SE* = 1.03, *t* = 3.12, *p* < 0.001) and higher (*B* = 6.22, *SE* = 1.03, *t* = 6.06, *p* < 0.001) emotional difficulties had better scores in the greenness than indoor (Figure 2).

For sustained attention, a similar LMM revealed no significant direct and interactive effects (Table 2, panel B).

### 3.2. RQ2: Effect of the Environment on Math Calculation Performance

For both calculation tasks, LMMs revealed again the main effect of the environment. Regarding the shorter-time and less difficult task (Table 3, panel A), the model explained 91% of the variance (marginal *R*^2^ = 4% explained by the fixed effects). Children had higher scores in the task with additions and subtractions after the green than the classroom lesson. For the longer-time and more difficult task, not only the environment but also emotional difficulties had effects on children’s performance (Table 3, panel B). Better scores were achieved when the task was performed in the greenness and by students with lower self-reported emotional difficulties. The model explained 81% of the variance (marginal *R*^2^ = 7% explained by the fixed effects). The interactive effect of environment and emotional difficulties did not emerge.

### 3.3. RQ3: Effect of the Environment on Affect

From a LMM for pre-post lesson positive affect the three-way interaction of time, environment, and emotional difficulties emerged (Table 4, panel A). The model explained 66% of the variance (marginal *R*^2^ = 7% explained by the fixed effects).

The slope analysis showed that students with higher emotional difficulties reported a statistically significant increase of positive affect after the school lesson in the greenness than after the lesson in the classroom (*B* = 0.97, *SE* = 0.37, *t* = 0.64, *p* = 0.01). In contrast, positive affect of students with lower emotional difficulties did not vary as a function of the environment where the lesson took place (*B* = −0.01, *SE* = 0.37, *t* = −0.04, *p* = 0.97). Figure 3 shows the results of the three-way interaction.

From a similar LMM for pre-post lesson negative affect, the two-way interaction between environment and emotional difficulties as well as the three-way interaction between time, environment, and emotional difficulties emerged (Table 4, panel B). The model explained 61% of the variance (marginal *R*^2^ = 6% explained by the fixed effects). The slope analysis for the three-way interaction revealed that students with higher emotional difficulties reported lower negative affect after the lesson in the green then after the lesson in the classroom, even though the difference between the two environments did not reach statistical significance (*B* = −0.08, *SE* = 0.04, *t* = −1.71, *p* = 0.09). For students with lower emotional difficulties, negative affect reported after the lesson did not vary as a function of the environment in which the lesson took place (*B* = −0.02, *SE* = 0.04, *t* = −0.48, *p* = 0.63). Figure 4 shows the results of the three-way interaction.

### 3.4. RQ4: Effect of the Environment on Perceived Restorativeness

A LMM model for perceived environmental restorativeness revealed the main effect of the environment where the lesson took place (Table 5). Children rated the outdoor green garden as much more restorative than the indoor classroom. The model explained 71% of the variance (marginal *R*^2^ = 8% explained by fixed effects).

## 4. Discussion

The study sought to extend current knowledge of the benefits of exposure to nature by investigating the cognitive and emotional effects of a single school lesson taught in the usual classroom and in the green school garden. To our knowledge this is the first study focused on a short and passive exposure to a natural environment for attending a single school lesson. In fact, previous research concerned cumulative, long-term effects of multiple outdoor lessons [9], and short-term passive experience with nature during green breaks [10].

The first research question asked whether attention scores differed according to the environment where the school lesson took place and whether students’ emotional difficulties would moderate the relationship between the environment and attentional performance. The findings partially confirmed our hypothesis (Hypothesis 1) as children’s selective attention was greater after the lesson in the greenness than in the classroom. This outcome suggests that after an exposure to nature not for a recess during a school day but for listening to a traditional lesson, young students are more able to focus their attention on a typical lab task than after a similar lesson in the usual classroom. In contrast, no statistically detectable differences emerged for sustained attention, although the scores were slightly higher after the outdoor lesson. Paying attention to the lesson in the greenness was less demanding in terms of attentional resources compared to the classroom lesson. Interestingly, children’s self-reported emotional difficulties moderated the relationship between the environment and attentional performance. Students with higher self-reported emotional difficulties were those who benefitted more from a lesson in the greenness, an outcome that suggests how a calmer and quieter context for a lesson supports selective attention [49].

Taken together the findings about attentional performance suggest that the positive effect of nature mainly reflects in the kind of attention that allows to be focused on a particular object and ignore irrelevant or distracting stimuli, that is, a kind of attention that has been frequently examined in previous studies on the benefits of nature (e.g., [30]). Our results not only provide further evidence of the attention restoration theory [12,14], but also expand current data by showing that even after a passive and short contact with a natural environment for listening to a lesson, not for resting [10], children have better selective or focused attention compared to their experience in the classroom environment, and those with higher emotional difficulties benefit more from the exposure to nature. This outcome also aligns with previous research showing the combination of cognitive and emotional benefits from contacts with nature [4]. Sustained attention was not related to the environment, likely because other factors that were not measured in the current study, may also be involved in the ability to maintain attention for a prolonged period of time.

The second research question asked whether children’s performance in typical school math tasks also differed in relation to the environment in which the lessons took place and whether their emotional difficulties would moderate the relationship. As hypothesized (Hypothesis 2) in both math tasks—the shorter-time task with the calculation of additions and subtractions and the longer-time task with the calculation of additions, subtractions, multiplications, and divisions—children performed better after the outdoor lesson in the greenness that in the classroom. Therefore, not only in lab attention tasks used in previous research [29,30], but also in usual school tasks, the positive cognitive impact emerges from a short and passive exposure to nature while listening to a teacher lesson. Not surprisingly, children’s emotional difficulties were a negative predictor of their performance in the more difficult math calculation task. The relationship between negative affect (e.g., anxiety) and math performance is well established in the literature e.g., [61].

The third research question asked whether children’s affective states differentiated according to the environment in which the lesson took place and whether children’s emotional difficulties moderate the relationship. Regarding positive affect, the main effect of the environment did not emerge, contrary to our hypothesis (Hypothesis 3). However, emotional difficulties moderated the relationship between the environment and the outcome variable as children with higher internalizing problems showed a significant increase in positive affect only in the green environment compared to those with less emotional difficulties.

Regarding negative affect, again, the environment per se did not have an effect and the moderating role of emotional difficulties did not emerge clearly. Students with higher difficulties in the affective domain tended to report a decrease in negative affect after the lesson in the greenness than in the classroom, but the difference between the two environments was not clearly detectable. Therefore, our Hypothesis 3 was not confirmed.

Our findings on affect are at least partially in line with those regarding the benefits of nature on health and wellbeing of people at various ages, including children [4,5]. Although we cannot state that being outside for the lesson reduced students’ stress with reference to the stress reduction theory [13,15], as we did not measure this variable, we can maintain that after a single lesson in nature, children with higher emotional difficulties reported more positive affect and a tendency to have lower negative affect.

The fourth and last research question asked whether students perceive differently the restorative quality of the physical environment where the lessons were taught and whether their emotional difficulties moderate such perception. As hypothesized (Hypothesis 4), children perceived more positively the green environment for its higher regenerative quality, an outcome that confirms previous research with young students [29,62].

### Limitations and Directions for Future Research

As for any study, the current findings should be interpreted in light of some limitations. First, we only considered emotional problems as a possible moderator of the relation between the environment and the outcome variables. With a larger sample other individual differences can be examined, for example inhibition ability that is involved in effortful control during school lessons. Second, the reliability of some scales is lower than desirable, but still acceptable for research purposes. Third, our data only regard young students in the second and third grades of primary school and cannot be generalized to other grade levels. Future studies need to compare cognitive and emotional effects of green environments on students at various school ages. Fourth, to keep the conditions as controllable as possible, we did not focus on new learning performance as an effect of exposure to nature. However, future studies can investigate the effects of the environment on the acquisition of new knowledge introduced to students in an outdoor and indoor lesson. Fifth, with this respect, although the lessons in the two environments were similar for difficulty according to the teachers’ judgments, we did not use a performance measure to indicate the equivalence of the lessons for students’ engagement and learning. Next investigation on the effects of regular lessons in the greenness should objectively document their equivalence in terms of cognitive demands to students.

## 5. Conclusions

Notwithstanding these limitations, the study has theoretical significance as it indicates the positive cognitive and emotional effects of passive exposure to a natural environment during a single school lesson, in particular for children with higher self-reported emotional difficulties. They may show better ability to focus on a task ignoring distractions and to experience more positive and less negative affect when they are exposed to a green environment even for listening to a teacher lesson. The associations of attentional skills and positive affective states with academic achievement are well established.

The study therefore also has practical significance as it is important to support children’s cognitive, academic, and emotional functioning even indirectly by giving them, even occasionally, the opportunity to spend time in nature not only for a recess but also for regular lessons. This becomes very important for today children who spend less and less time outdoor and many have very little contact with nature. The physical environment where learning activities take place matters. Especially, if the weather permits, for example once a week or more occasionally, or when following demanding lessons regarding particularly complex topics, utilizing the green school courtyard or a near park for teaching can be feasible, easy, and relatively at a low cost. Even a single lesson supports children’s cognitive performance and affect in subsequent tasks, in particular for those who are emotionally more fragile and can experience a calmer and quieter environment outside the classroom, in the greenness.

## Figures and Tables

**Figure 1 ijerph-19-16823-f001:**
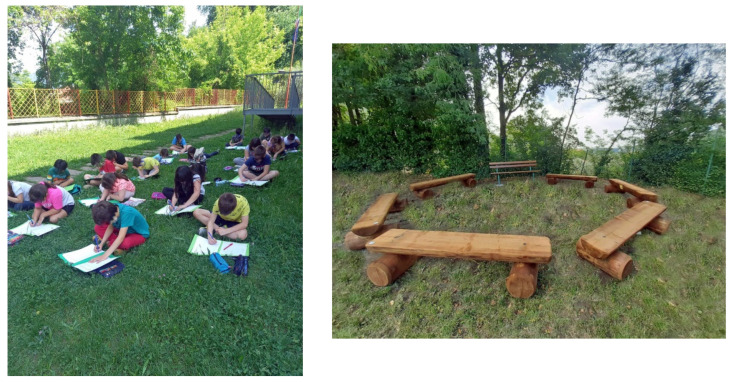
The natural environment of the two schools.

**Figure 2 ijerph-19-16823-f002:**
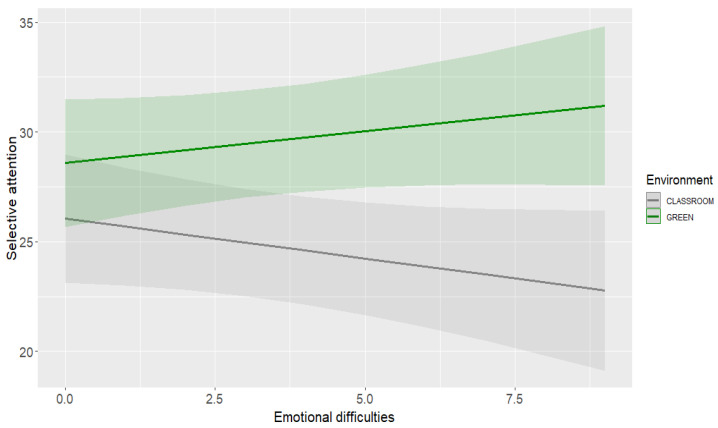
Interaction effect of environment and emotional difficulties on selective attention.

**Figure 3 ijerph-19-16823-f003:**
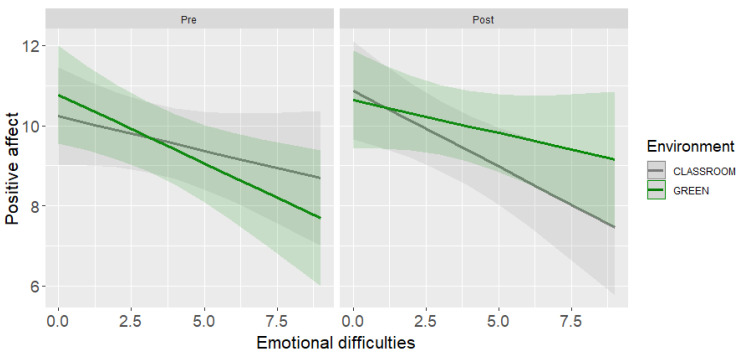
Interaction effect of environment, pre-post positive affect, and emotional difficulties.

**Figure 4 ijerph-19-16823-f004:**
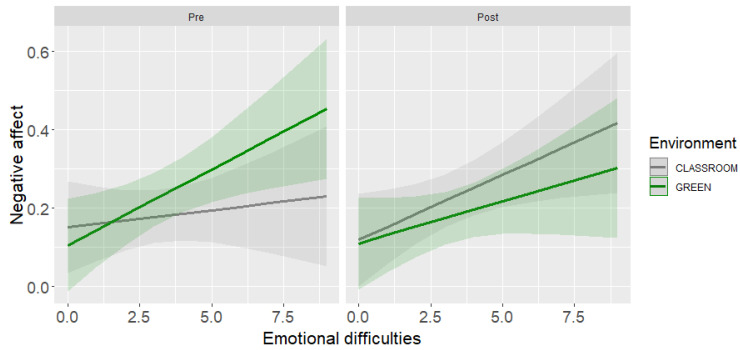
Interaction effect of environment, pre-post negative affect, and emotional difficulties.

**Table 1 ijerph-19-16823-t001:** Descriptive statistics for all variables for the entire sample and by environment (N = 65).

	*M* (*SD*)	*Skewness* (*SE*)	*Kurtosis* (*SE*)
Emotional problems	3.32 (2.31)	0.28 (0.29)	−0.94 (0.58)
Classroom			
Selective attention	24.94 (5.31)	−0.47 (0.29)	0.66 (0.58)
Sustained attention	30.28 (3.61)	−0.28 (0.29)	−2.28 (0.58)
Shorter-time calculation	24.45 (10.96)	0.41 (0.29)	0.26 (0.58)
Longer-time calculation	4.45 (2.47)	−1.13 (0.29)	−1.15 (0.58)
Positive affect-pre	9.71 (2.16)	−0.94 (0.29)	−1.30 (0.58)
Positive affect-post	9.66 (2.61)	−1.25 (0.29)	1.23 (0.58)
Negative affect-pre *	0.89 (1.60)	1.25 (0.29)	0.50 (0.58)
Negative affect-post *	1.25 (2.09)	1.06 (0.29)	0.20 (0.58)
Perceived restorativeness	23.65 (10.61)	−0.24 (0.29)	−9.77 (0.58)
Greenness			
Selective attention	29.65 (3.29)	0.93 (0.29)	0.30 (0.58)
Sustained attention	31.22 (3.69)	−1.39 (0.29)	−0.63 (0.58)
Shorter-time calculation	26.74 (13.18)	−0.80 (0.29)	0.26 (0.58)
Longer-time calculation	4.92 (2.33)	−0.33 (0.29)	−0.81 (0.58)
Positive affect-pre	9.68 (2.88)	−1.43 (0.29)	1.53 (0.58)
Positive affect-post	10.14 (2.37)	−1.24 (0.29)	1.00 (0.58)
Negative affect-pre *	1.26 (1.99)	0.94 (0.29)	−0.44 (0.58)
Negative affect-post *	0.85 (1.38)	1.05 (0.29)	−0.19 (0.58)
Perceived restorativeness	29.74 (8.78)	−0.23 (0.29)	−1.61 (0.58)

Note. * = after log-transformation.

**Table 2 ijerph-19-16823-t002:** Linear mixed models for selective and sustained attention.

	Panel A				Panel B		
	Selective Attention				Sustained Attention		
*Predictors*	*B*	95% *CI*	*p*	*Predictors*	*B*	95% *CI*	*p*
(Intercept)	26.05	23.11–28.99	<0.001	(Intercept)	29.63	27.20–32.05	<0.001
Environment [Green]	2.53	0.01–5.05	0.049	Environment [Green]	0.92	−0.41–2.24	0.173
Emotional difficulties	−0.36	−0.85–0.12	0.138	Emotional difficulties	0.19	−0.19–0.56	0.319
Envir. [Green] * Emot. diff.	0.65	0.03–1.28	0.040	Envir. [Green] * Emot. Diff.	0.01	−0.32–0.33	0.968
**Random Effects**				**Random Effects**			
σ^2^	17.05			σ^2^	4.68		
τ_00 ID_	1.86			τ_00 ID_	6.32		
τ_00 Class_	5.01			τ_00 Class_	3.73		
ICC	0.29			ICC	0.68		
Marg. *R*^2^/Cond. *R*^2^ = 0.205/0.433				Marg. *R*^2^/Cond. *R*^2^ = 0.028/0.691			

**Table 3 ijerph-19-16823-t003:** Linear mixed models for shorter and longer-time math calculation tasks.

	Panel A				Panel B		
	Shorter-Time Calculation				Longer-Time Math Calculation		
*Predictors*	*B*	95% *CI*	*p*	*Predictors*	*B*	95% *CI*	*p*
(Intercept)	26.38	17.94–34.81	<0.001	(Intercept)	5.01	3.24–6.77	<0.001
Environment [Green]	2.79	0.58–5.00	0.014	Environment [Green]	0.71	0.07–1.35	0.029
Emotional difficulties	−0.82	−1.94–0.30	0.149	Emotional difficulties	−0.22	−0.40–−0.03	0.023
Envir. [Green] * Emot. diff.	−0.15	−0.70–0.40	0.590	Envir. [Green] * emot. diff.	−0.07	−0.23–0.09	0.380
**Random Effects**				**Random Effects**			
σ^2^	13.06			σ^2^	1.08		
τ_00 ID_	82.77			τ_00 ID_	1.60		
τ_00 Class_	52.42			τ_00 Class_	2.61		
ICC	0.91			ICC	0.80		
Marg. *R*^2^/Cond. *R*^2^ = 0.036/0.915				Marg. *R*^2^/Cond. *R*^2^ = 0.070/0.810			

**Table 4 ijerph-19-16823-t004:** Linear mixed models for pre-post positive and negative affect.

	Panel A				Panel B		
	Positive Affect				Negative Affect		
*Predictors*	*B*	95% *CI*	*p*	*Predictors*	*B*	95% *CI*	*p*
(Intercept)	10.23	9.01–11.46	<0.001	(Intercept)	0.15	0.03–0.27	0.012
Environment [Green]	0.54	−0.36–1.43	0.240	Environment [Green]	−0.05	−0.15–0.06	0.399
Pre-post [Post]	0.64	−0.25–1.54	0.159	Pre-post [Post]	−0.03	−0.14–0.08	0.553
Emotional difficulties	−0.17	−0.43–0.09	0.190	SDQ Emotional difficulties	0.01	−0.02–0.04	0.561
Envir. [Green] * Pre-post [Post]	−0.77	−2.04–0.50	0.234	Envir. [Green] * Pre-post [Post]	0.04	−0.12–0.19	0.632
Envir. [Green] * Emot. diff.	−0.17	−0.39–0.05	0.132	Envir. [Green] * Emot. diff.	0.03	0.00–0.06	0.028
Pre-post [Post] * Emot. diff.	−0.21	−0.43–0.01	0.067	Pre-post [Post] * Emot. diff.	0.02	−0.00–0.05	0.072
(Envir. [Green] * Pre-post [Post]) * Emot. diff.	0.38	0.07–0.70	0.017	(Envir. [Green] * Pre-post [Post]) * Emot. diff.	−0.04	−0.08–−0.00	0.030
**Random Effects**				**Random Effects**			
σ^2^	2.17			σ^2^	0.03		
τ_00 ID_	3.32			τ_00 ID_	0.04		
τ_00 Class_	0.44			ICC	0.58		
ICC	0.63						
Marg. *R*^2^/Cond. *R*^2^ = 0.072/0.660				Marg. *R*^2^/Cond. *R*^2^ = 0.060/0.606			

**Table 5 ijerph-19-16823-t005:** Linear mixed for perception of environmental restorativeness.

	Perceived Restorativeness		
*Predictors*	*B*	95% *CI*	*p*
(Intercept)	23.13	16.50–29.76	<0.001
Environment [Green]	6.83	3.31–10.35	<0.001
Emotional difficulties	0.17	−0.81–1.15	0.728
Envir. [Green] * Emot. diff.	−0.22	−1.09–0.65	0.615
**Random Effects**			
σ^2^	33.21		
τ_00 ID_	42.01		
τ_00 Class_	29.28		
ICC	0.68		
Marg. *R*^2^/Cond. *R*^2^ = 0.083/0.709			

## Data Availability

The dataset generated during and/or analyzed during the current study is available from the corresponding author upon reasonable request.

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
