# Peer review of "Lessons in a Green School Environment and in the Classroom: Effects on Students’ Cognitive Functioning and Affect"

_ijerph, 2022, doi:10.3390/ijerph192416823_

Round 1

Reviewer 1 Report

This manuscript discusses how exposure to a green environment impacts/benefits learners, as opposed to them learning in a conventional classroom. The literature review presented is organised and informs the reader adequately the purpose, though one of the gaps mentioned (p.5, line 168) should be verified and rephrased. If no studies have really been undertaken, it could be an indication of the worthiness (or lack of) of this study.

The measures used in this study appear appropriate. For example, the PANAS cited suggests that the positive and negative affects could be presented regardless of sample (i.e., not heavily reliant on sample characteristics). Nonetheless, this might not be said for the SDQ, that was validated based on a sample comprising psychiatric and dental clinic attenders. In this regard, the authors could consider inserting a short para for each of the measures to justify that other studies with similar sample characteristics have also used these measures; this would convince readers that the measures used, and hence the results are adequately robust.

One other minor suggestion is to append a picture of the green outdoor space. This might interest readers (and benefit the journal) in that learning environment designers could appreciate how the "green" space looked like in this study.

Overall, this manuscript was presented in a way that a reader could follow without issue. Thank you.

Author Response

To Reviewer #1

In our response letter, we answer each of your comments and recommendations. We reiterate each of the issues raised (in italics) and detail how we have addressed each issue. Page numbers are provided. All revisions are tracked in the text to facilitate their identification.

This manuscript discusses how exposure to a green environment impacts/benefits learners, as opposed to them learning in a conventional classroom. The literature review presented is organised and informs the reader adequately the purpose, though one of the gaps mentioned (p.5, line 168) should be verified and rephrased.  If no studies have really been undertaken, it could be an indication of the worthiness (or lack of) of this study.

RESPONSE. Thank you for your comment. We mean to say that there are no studies – to our knowledge – that investigated the effects of a single lesson in the greenness. We already know the positive effects of a series of lessons over an extended period of time or the positive effects of short green breaks (15-20 minutes) during a school day (as reported on p. 2). However, we do not know whether a single lesson (around 1 hour) in a green environment, which requires to pay attention to what the teachers says, is beneficial for both cognition and affect. We have revised some sentences according to your comment  (p. 5).   

The measures used in this study appear appropriate. For example, the PANAS cited suggests that the positive and negative affects could be presented regardless of sample (i.e., not heavily reliant on sample characteristics). Nonetheless, this might not be said for the SDQ, that was validated based on a sample comprising psychiatric and dental clinic attenders. In this regard, the authors could consider inserting a short para for each of the measures to justify that other studies with similar sample characteristics have also used these measures; this would convince readers that the measures used, and hence the results are adequately robust.

RESPONSE.  We  have followed your recommendation and have added important details (p. 7). Yes, the SDQ was originally validated on a sample of parents and teachers of children drawn from dental and psychiatric clinics. The child version has also been used with samples of typically developing fifth graders (Scrimin et al., 2015). Based on our participants’ literacy and comprehension skills, we used with younger students the child version of the scale included in the SDQ. Importantly, each question was read aloud to the class, and any difficult words or phrases were explained and answers were provided when children asked clarifications . Please consider that we could not ask participants’ teachers to fill in the teacher SDQ version for privacy regulations.

One other minor suggestion is to append a picture of the green outdoor space. This might interest readers (and benefit the journal) in that learning environment designers could appreciate how the "green" space looked like in this study.

RESPONSE. Great suggestion: we have inserted a picture of the green environment  of the two schools, which is now Figure 1. (p. 8).

Overall, this manuscript was presented in a way that a reader could follow without issue. Thank you.

RESPONSE.  We thank you for your positive evaluation and constructive comments to improve the quality of our work.  

Reviewer 2 Report

·         General comments:

The proposal falls within the theme of the journal and generally complies with the necessary standards to publish in it in terms of both formal and content aspects.

Although the topic is not innovative, the perspective on emotions in students with more difficulties and the relationship between positive affective states and academic achievement in the classroom is innovative.

It is surprising to observe that with just a single session, improvement can be appreciated -saying this not as a negative criticism of the work.

·         Introduction

There are studies that show that being in nature is distracting, etc.; scientific literature that does not support the line of research of the article must also be collected for its objectives.

In line 122, mentions to the references “Bustos Ibarra, A. V., Montenegro Villalobos, C. S., & Batista Kida, A. de S. (2021). Use of teaching oral regulation as a strategy to help reading comprehension: an experimental study in third grade students of chilean schools. Investigaciones Sobre Lectura, (15), 64-94. https://doi.org/10.24310/isl.vi15.12562 ” and “Jiménez-Pérez, E., Martínez-León, N., & Cuadros-Muñoz, R. (2020). Maternal influence on the emotional intelligence and reading competence of their children. Ocnos, 19 (1), 80-89.doi: HTTPS://DOI.ORG/10.18239/OCNOS_2020.19.1.2187” could also endorse the importance of being focused when reading and writing.

In line 163, mentions to the reference “Jiménez-Pérez, E.; de Vicente-Yagüe Jara, M.-I.; Gutiérrez-Fresneda, R.; García-Guirao, P. Sustainable Education, Emotional Intelligence and Mother–Child Reading Competencies within Multiple Mediation Models. Sustainability, 13, 1803. https://doi.org/10.3390/su13041803 could also endorse the importance of emotions in school.

·         Method

The aspects that were intended to be analyzed from a statistical and formal point of view have been considered in a solid and solvent way. There is nothing more to add.

·         Conclusion

If you truly want to transfer knowledge, you should specify how this contact with nature should take place in the classroom; only by providing specific suggestions can educational centers pick them up. Perhaps teach a different subject every week in the nearest park? It is possible to be objective while avoiding indoctrination without sacrificing ambition in the conclusions. 

Author Response

To Reviewer # 2

In our response letter, we answer each of your comments and recommendations. We reiterate each of the issues raised (in italics) and detail how we have addressed each issue. Page numbers are provided. All revisions are tracked in the text to facilitate their identification.

General comments:

The proposal falls within the theme of the journal and generally complies with the necessary standards to publish in it in terms of both formal and content aspects.

RESPONSE. Thank your overall positive comment.

Although the topic is not innovative, the perspective on emotions in students with more difficulties and the relationship between positive affective states and academic achievement in the classroom is innovative.

RESPONSE. We also thank you for this more specific positive comment

It is surprising to observe that with just a single session, improvement can be appreciated -saying this not as a negative criticism of the work.

RESPONSE. We are glad for your appreciation. Yes, we were interested in examining whether a single lesson taught in the greenness would have better cognitive and affect impact than a single comparable lesson taught in the classroom  to the same children. It is more predictable that a series of lessons in a natural environment have benefits, but it is also relevant, both theoretically and practically, to document that even a single lesson is beneficial for students’ cognitive and affective functioning. 

Introduction

There are studies that show that being in nature is distracting, etc.; scientific literature that does not support the line of research of the article must also be collected for its objectives.

RESPONSE.  Yes, there are also mixed results in the literature, especially in research addressing the relationship between green cover  in some areas and students’ achievement.  Moreover, Mygind et al. (2018) also reported non cognitive benefits in primary-school children after short rest in nature. We have elaborated on the issue and added relevant references (pp. 3-4).

In line 122, mentions to the references “Bustos Ibarra, A. V., Montenegro Villalobos, C. S., & Batista Kida, A. de S. (2021). Use of teaching oral regulation as a strategy to help reading comprehension: an experimental study in third grade students of chilean schools. Investigaciones Sobre Lectura, (15), 64-94. https://doi.org/10.24310/isl.vi15.12562 ” and “Jiménez-Pérez, E., Martínez-León, N., & Cuadros-Muñoz, R. (2020). Maternal influence on the emotional intelligence and reading competence of their children. Ocnos, 19 (1), 80-89.doi: HTTPS://DOI.ORG/10.18239/OCNOS_2020.19.1.2187” could also endorse the importance of being focused when reading and writing.

RESPONSE. Thank your for your suggestions. We have included the reference to Bustos Ibarra et al. (2021) as more pertinent than that by Jiménez-Pérez et al. (2020), which does not refer to the school context. 

In line 163, mentions to the reference “Jiménez-Pérez, E.; de Vicente-Yagüe Jara, M.-I.; Gutiérrez-Fresneda, R.; García-Guirao, P. Sustainable Education, Emotional Intelligence and Mother–Child Reading Competencies within Multiple Mediation Models. Sustainability, 13, 1803. https://doi.org/10.3390/su13041803 could also endorse the importance of emotions in school.

RESPONSE.  We have not added this reference as it is not really relevant and in line with what is argued at that point of the manuscript. More precisely, in the study published in Sustainability  mothers’ emotional intelligence were examined, not students’ (or teachers’) emotions.  I hope you can understand the reason.

Method

The aspects that were intended to be analyzed from a statistical and formal point of view have been considered in a solid and solvent way. There is nothing more to add.

RESPONSE. Thank you for your positive comment.

Conclusion

If you truly want to transfer knowledge, you should specify how this contact with nature should take place in the classroom; only by providing specific suggestions can educational centers pick them up. Perhaps teach a different subject every week in the nearest park? It is possible to be objective while avoiding indoctrination without sacrificing ambition in the conclusions.

RESPONSE. We understand well your point. We have added specific suggestions for educational practice (p. 16).

We thank you for your positive evaluation and constructive comments to improve the quality of our work.  
